# Geopolitical Risks and Yield Dynamics in the Australian Sovereign Bond Market

Milan Christian De Wet

Department of Accountancy, College of Business and Economics, University of Johannesburg, Auckland Park 2006, South Africa; miland@uj.ac.za

**Abstract:** Geopolitical risks and shocks such as military conflicts, terrorist attacks, and war tensions are known to cause significant economic downturns. The main purpose of this paper is to determine the dynamics between Australian sovereign bond yields and geopolitical risk. This is achieved by employing a quantile regression analysis. The findings of this study indicate that the impact of geopolitical risk on Australian sovereign yield dynamics is asymmetrical. Furthermore, an increase in geopolitical risk only impacts short-term yields at extreme regimes. However, the impact is, by and large, insignificant. On the other hand, an increase in geopolitical risk does have a statistically significant positive impact on medium- and long-term yields across most quantiles. Lastly, an increase in geopolitical risk tends to result in a steeper yield curve at the belly of the curve but causes the yield curve to flatten at the long end. This study is the first study that holistically examines the dynamics between geopolitical risk and Australian sovereign bond yields. The study thereby contributes to the body of knowledge on Australian bond yields, specifically, and adds to the sparse body of knowledge on the dynamics between geopolitical risk and sovereign bond yields. The findings of this study have implications for monetary policy makers, given that shifts in sovereign bond yields could impact all three core mandates of the Australian Reserve Bank. Furthermore, changes in the slope of the yieldcurve could be used by monetary policy makers to pre-empt changes in future economic growth. The results of this study also relate to fiscal policy formulation, given that yields directly impact the cost of government borrowing. Lastly, portfolio managers could benefit from the results of this study, as these results provide information on the ability of Australian sovereign bonds to hedge against geopolitical risk.

**Keywords:** geopolitical risk; Australian bond yields; quantile regression; yield curve

## 1. Introduction

Geopolitical risks and shocks such as military conflicts, terrorist attacks and war tensions have proved to have an impact on both economic activity and financial markets (Glick and Taylor 2010). Furthermore, geopolitical risks are identified by central banks and many market participants as a significant driver of uncertainty and by implication, a determinant of investment decision-making (Caldara and Iacoviello 2022). It is well documented in literature that an increase in uncertainty results in a risk premium. Therefore, through the investment channel, geopolitical risk could have an impact on financial markets, financial stability, and the business cycle. Furthermore, recent literature suggests that geopolitical tensions play an increasingly important role in shaping financial interactions at a multinational level (Dogan et al. 2021). In this light, central banks and market participants are increasingly considering the impact of geopolitical risk on returns and volatility of financial assets. Evidence by Caldara and Iacoviello (2022) shows that increased levels of geopolitical risk cause a reduction in economic activity and a flow of capital from relatively high-risk assets to relatively low-risk assets. This results in a general decline in stock returns, and a flow of capital out of developing countries as well as developed countries with considerable international exposure (Caldara and Iacoviello 2022).

The re-allocation of capital from economies that are more exposed to geopolitical risk to economies with lower exposure brings sovereign bond yields to the fore. Sovereign bonds play a vital role in the financial health of an economy, and the ability of governments to access the capital markets at a reasonable cost of capital is essential to the stability of the local economy (Le and Tran 2021). Given the significant role that sovereign bonds play within an economy, it is important to understand the impact of the drivers thereof. Yet, the impacts of an increase in geopolitical risk on sovereign bond yields are unclear and depend on the risk profile of the sovereign bond. On the one hand, some sovereign bonds are considered by the market to be safe haven assets, and an increase in geopolitical risk could lead to a decline in the yields of such bonds. For example, Treasury bills and bonds issued by the government of the United States of America (US) have proven to act as safe havens against heightened risk factors (Baur and McDermott 2016).

On the other hand, sovereign bonds issued by governments with fiscal revenues negatively exposed to geopolitical tensions could attract a risk premium, resulting in an increase in yields. This, in turn, could have an impact on micro-investment decisions, asset pricing and general economic activities. Le and Tran (2021) purport that a deterioration in financing conditions could be the transmission channel through which geopolitical risk impacts the local economy. Evidence of this is provided by Caldara and Iacoviello (2022), signalling that heightened geopolitical risk typically causes sovereign bond yields of emerging economies to increase. In part, this is reflected by a typical increase in sovereign quality spreads during periods of elevated geopolitical risk (Baldacci et al. 2011). Furthermore, research shows that geopolitical tensions have a significant impact on term spreads, also known as curve spreads, thus influencing yield curve dynamics (Subramaniam 2022).

Research by Subramaniam (2022) additionally reveals that the impact of geopolitical risk on bond yields depends on the prevailing monetary policy regime. Therefore, geopolitical risk has an asymmetric impact on yields, depending on the prevailing rate cycle (De Wet and Botha 2022). For example, Sohag et al. (2022) have found that yields on medium-dated bonds tend to respond positively to an increase in geopolitical tension during relatively low-interest regimes, that is, at the lower 10th to 40th quantiles. Yet Sohag et al. (2022) found that yields on the same bonds also tend to respond negatively to an increase in geopolitical tension during relatively high-interest regimes, that is, at the upper 80th to 90th quantiles. This complicates the analysis of the dynamics between sovereign yields and geopolitical risk and standard linear modelling will not provide an accurate estimation of this dynamic.

The above discussion suggests that the impacts of geopolitical risk on sovereign bond yields are complex, and likely to be heterogeneous across bond markets and rate cycles. This, in combination with the reduced effectiveness of monetary policy during heightened political tension, as shown by Marfatia (2015), renders monetary policy formulation particularly challenging during these periods. A good starting point to manage the spillover risk posed by geopolitical tensions is to identify the impact of geopolitical tension on sovereign bond yields. However, despite some recent research to this end (see, for example, Gupta et al. (2021)), research remains very limited. Most research focuses on political risk, evidenced for example by Afonso et al. (2015) and Bianchi et al. (2017). However, given the distinctly different nature of geopolitical risk, and the importance of government bond yields to the financial stability of a local economy, there is room for the body of knowledge on how geopolitical risk impacts bond yields to be expanded.

The importance of addressing this research gap is amplified by the heterogeneous nature of bond markets across countries. This heterogeneity is caused by the heterogeneous nature of the bond issuer—in this case, the government. The fiscal position of a government, underpinned by economic activity, is an important factor in the ability of a government to meet its debt obligations. In turn, the nature and composition of economic activities could be vastly different from one country to the next, and thus, the impact of geopolitical risk on the local economy could differ. As a result, the risk premium required by bond investors in the face of elevated geopolitical risk could differ significantly from one country to the next.

Caldara and Iacoviello (2022) substantiate this, arguing that the way in which markets respond to geopolitical events varies across economies. This requires multiple studies in this field, focusing on various countries, to obtain a holistic understanding of the impact of geopolitical risk on sovereign bond yields.

To date, no study has been conducted to test the dynamics between geopolitical risk and the Australian bond market empirically. The Australian bond market is of particular interest in this case because there are two conflicting elements at play during heightened geopolitical risk periods. On the one hand, Australian sovereign bonds are considered high-quality, low-risk debt instruments. This is indicated by the fact that all three major global credit rating agencies rate Australian sovereign credit as AAA, the highest sovereign credit rating attainable. This is higher than the AA+ allocated by Standard and Poor for US sovereign debt. It suggests that during periods of uncertainty, Australian sovereign bonds could act as a safe haven. On the other hand, the Australian economy is considered an open economy, with 22.2% of Gross Domestic Product (GDP) derived from exports in 2021 (World Bank 2022). In 2021, iron ore constituted approximately 31.8% of exports, and commodities overall constituted more than 60% of exports (World Bank 2022). Additionally, 40.7% of Australian exports are to China; Balcilar et al. (2018) provide evidence that the Chinese economy is particularly exposed to geopolitical risk.

Furthermore, if increased geopolitical tensions result in a global economic slowdown, the commodity cycle could contract and negatively impact Australian economic growth. The question is thus which element dominates investment decision-making, and subsequently, how Australian sovereign bonds perform during heightened geopolitical risk periods. In this light, the study aims to determine the dynamics between the Australian sovereign bond market and geopolitical risk.

This study aims to test three hypotheses:

**Hypothesis 1 (H1).** *The first hypothesis directly relates to the impact of geopolitical risk on the yield of short-, medium-, and long-dated Australian sovereign bonds. The null hypothesis is that an increase in geopolitical risk causes yields to rise. The alternative hypothesis is that an increase in geopolitical risk causes yields to contract.*

**Hypothesis 2 (H2).** *The second hypothesis relates to the impact of geopolitical risk on yield curve dynamics in the Australian bond market. The null hypothesis is that an increase in geopolitical risk causes the yield curve to flatten. The alternative hypothesis is that an increase in geopolitical risk causes the yield curve to steepen.*

**Hypothesis 3 (H3).** *The third hypothesis relates to the impact of geopolitical risk on quality spreads, specifically focusing on the spread between US and Australian sovereign bond yields of a similar maturity. The null hypothesis is that an increase in geopolitical risk causes quality spreads to expand. The alternative hypothesis is that an increase in geopolitical risk causes quality spreads to contract.*

To account for the asymmetries identified by Sohag et al. (2022) and Subramaniam (2022) regarding the dynamic between geopolitical risk and sovereign yields, this study employs a quantile regression analysis. This is a commonly used model in literature to capture asymmetric relationships by analysing variables at different quantiles (Subramaniam 2022). This will allow one to account for different interest rate regimes and spread levels, thus capturing the dynamics between yields and geopolitical tensions under various market conditions. Subsequently, the above-mentioned hypotheses will be tested at various quantile intervals.

This study contributes to the body of empirical knowledge on the dynamics between geopolitical risk and bond yields, and is the first study that examines the dynamics between geopolitical risk and the Australian sovereign bond market. The study does so by considering yields related to different maturities, as well as considering term spreads and quality spreads. All the variables are considered at a range of quantiles. This is particularly useful with yield spreads as a means of gauging the yield curve dynamics, given that yield spread

sensitivity is largely influenced by the prevailing steepness of the curve. Information on the dynamics between geopolitical risk and sovereign bond yields at a quantile level will offer policy makers and portfolio managers a base that can be used to improve policy accuracy and tactical asset allocation efficiency. The results identified in this study pertaining to short-dated bonds are relevant to monetary policy makers, given that monetary policy typically aims to control these rates. Thus, the results of this study will contribute towards monetary policy formulation in the face of geopolitical tensions, in relation to the prevailing interest rate conditions. Furthermore, medium- to long-dated bond yields have an impact on all three mandates of the Reserve Bank of Australia (RBA), since these rates could impact price stability, economic growth and employment. These results will therefore be of particular relevance to the RBA. Additionally, medium- to long-dated bond yields are typically determined by investor preferences, and information on the impact of geopolitical risk on these bonds will contribute towards effective portfolio construction and tactical asset allocation. These results will also contribute towards fiscal budgeting during periods of geopolitical risk, as these results provide knowledge of the impact of such risk on the cost of new government debt.

Results related to the impact of geopolitical risk on yield curve dynamics contribute by providing information on investors' risk sentiment. These, in turn, could be used to gauge expectations of future economic growth and could assist portfolio managers in effective tactical asset allocation. The results of this study also contribute towards the formulation of a hedging strategy against geopolitical risk. High-quality sovereign bonds are often utilised in portfolios as a means to hedge against risks that cause a decline in the risky assets (Baur and McDermott 2016). The idea is that high-quality sovereign bonds should appreciate during risk-adverse periods, and thus, yields of these assets should decline. The results of this study will provide empirical information that testify that Australian sovereign bonds could be considered a hedging option against geopolitical risk.

## 2. Literature Review

Spikes in geopolitical risk events have captured the attention of policy makers and academic researchers over the last decade. It is increasingly apparent that market participants consider geopolitical tensions as a risk factor, and that, therefore, such risks are discounted into in the pricing of financial assets (Subramaniam 2022. Geopolitical risk is not a new phenomenon, but the growing importance of geopolitical risks as a driving factor of financial asset prices over recent decades is largely a function of increasingly globally integrated economies and financial markets (Gupta et al. 2019; De Wet 2021). The integrated nature of economies and financial markets means that an escalation in geopolitical risk anywhere in the world will likely have a spillover effect on financial assets across economies and markets. In this light, the body of knowledge consists of several strands of research. The first relates to the impact of geopolitical risk on economic activity, including research on international trade; the second relates to the role of geopolitical risk in investment decision-making, including both fixed capital formation investments and portfolio investments; the third relates to the dynamics between geopolitical risk and financial markets. Literature on the latter is further split into research focusing on stock markets, commodity markets, the currency market and bond markets. The themes mentioned above are, however, interrelated by means of transmission channels, and cannot be viewed in isolation. This review of the literature will highlight those links.

Literature on economic activity and geopolitical risk includes research by Bloom (2009), Glick and Taylor (2010), Nikkinen and Vähämaa (2010), Caldara and Iacoviello (2022) and Gupta et al. (2019) Empirical results from these studies provide evidence that prolonged geopolitical tensions cause a slowdown in economic activity and could, depending on the severity, cause individual economies and the global economy to contract. Evidence from Gupta et al. (2019) advances that elevated geopolitical risk reduces output in emerging economies by between 13 and 22%. This slowdown feeds through various transmission channels. One channel is the consumption channel. Nikkinen and Vähämaa

(2010) provide evidence that prolonged geopolitical tensions cause consumers to postpone the consumption of non-essential goods and services due to uncertainty. Another channel is the international trade channel. Gupta et al. (2019) provide evidence that increased geopolitical risks have a negative impact on trade flows. Another channel through which geopolitical risk impacts the economy is the investment channel. This links with the body of literature on the role of geopolitical risk in investment decision-making.

Literature on investment decision-making and geopolitical risk includes research by Demir et al. (2019), Kotcharin and Maneenop (2020), Wang et al. (2020), Dogan et al. (2021), Elsayed and Helmi (2021), Le and Tran (2021), and Sohag et al. (2022). Findings by these researchers broadly concur that geopolitical risk is a key determinant in investment decision-making from both a fixed capital formation investment perspective and a portfolio investment perspective. Alterations in investors' future expectations are a key channel through which decisions are influenced (Sohag et al. 2022). In a similar light, increased uncertainty is identified in the literature as a channel through which investment decisions are altered during periods of high levels of geopolitical risk. Wang et al. (2020) argue that uncertainties caused by geopolitical risk typically result in significant risk aversion by investors.

In this light, Le and Tran (2021) provide evidence that geopolitical shocks have a significant negative impact on corporate investments, and thereby reduce fixed capital formation investments. The findings by Demir et al. (2019) corroborate this, showing that due to uncertainties, increased geopolitical risk causes friction in corporate policies. This, in turn, causes corporate entities to hold back on fixed capital formation investments, and firms tend to increase their liquidity position during periods of uncertainty. Kotcharin and Maneenop (2020) also demonstrate that firms tend to increase their cash holdings during periods of geopolitical uncertainty, to prepare for unexpected events. The findings by Hu and Gong (2019) are also important to note in this regard. They provide evidence that banks tend to reduce their lending activities to small and medium firms to reduce their exposure to default risk. This, in turn, could also restrict fixed capital formation activities. As mentioned before, the reduction in investment activities act as a transmission channel through which geopolitical risk impacts real economic activities.

Similar to fixed capital formation investing research, most literature provides evidence that geopolitical risk alters portfolio investment decision-making. This evidence indicates that investors tend to underweigh financial assets that are considered relatively risky and prefer capital preservation above returns during periods of geopolitical uncertainties (Kannadhasan and Das 2020; Wang et al. 2020). This re-allocation causes geopolitical risk to have an impact on financial market returns. It naturally leads to the third strand of literature related to the dynamics between geopolitical risk and financial markets. A number of researchers have considered the impact of geopolitical risk on the stock market—see, for example, Antonakakis et al. (2017), Balcilar et al. (2018), Gkillas et al. (2018), and Kannadhasan and Das (2020). Evidence provided by these researchers generally indicates that geopolitical risk has a negative impact on stock markets. Importantly, Kannadhasan and Das (2020) provide evidence of asymmetries, signalling that stock markets with values in the mediate and upper quantiles are more negatively impacted by an increase in geopolitical risk. Furthermore, Balcilar et al. (2018) suggest that spikes in geopolitical risk cause stock market volatility to increase significantly.

The above-referenced results are expected, given the inherently risky nature of stocks. However, formulating expectations around the dynamics between geopolitical risk and sovereign bond yields is more challenging. This is due to some sovereign bonds being considered to be safe-haven assets, owing to their lowrisk profile. On the other hand, sovereign bonds issued by governments with a weak underlying economy are typically considered to be risky assets (Favero et al. 2010). The drivers of sovereign bond yields and bond yield spreads have been widely considered in literature—see, for example, Favero et al. (2010), Aristei and Martelli (2014), Poghosyan (2014), and Afonso et al. (2015). Literature broadly provides evidence that monetary policy is a key driver of short to

medium sovereign bond yields, as well as a key determinant of term spreads (Poghosyan 2014; Bianchi et al. 2017). Literature also shows that yields and term spreads are significantly impacted by the fiscal accounts of the issuing government, macroeconomic conditions and international financial risks (Afonso et al. 2015). Furthermore, Bianchi et al. (2017) found sovereign credit ratings to have a significant impact on yields. The study by Bianchi et al. (2017) focused specifically on the Australian bond market and provides evidence that Australian sovereign bond yields are driven by similar factors, as identified by the broader body of literature.

Interestingly, Poghosyan (2014) provides evidence that the relationship between sovereign bond yields and their macroeconomic and fiscal determinants diminishes during periods of heightened risk and crisis, due to safe-haven flows. Literature related to the impact of risk on sovereign bond yields largely focuses on political risk and financial market risk—see, for example, Favero et al. (2010), Afonso et al. (2015), Aristei and Martelli (2014), and Poghosyan (2014). Despite the well-developed body of knowledge on sovereign bond yields and bond yield spreads, literature measuring the impact of geopolitical risk on sovereign bond dynamics is limited to only a few studies, including the work by Jalkh and Bouri (2022), Subramaniam (2022), and Gupta et al. (2021).

Gupta et al. (2021) completed their study on the US sovereign bond market. They provide evidence that geopolitical risk does have a causal effect on sovereign bond yields, with medium- to long-dated bonds being affected more severely than short-dated bonds. Similarly, Jalkh and Bouri (2022) considered the impact of geopolitical risk on the return and volatility of US Treasuries. The results by Jalkh and Bouri (2022) show that geopolitical risk does tend to cause US sovereign bond yields to decline; however, volatility in these securities tends to increase. The work by Subramaniam (2022) focuses on the yields of emerging market sovereign bonds. Subramaniam (2022) provides evidence that during a high-yield regime period, an increase in geopolitical risk causes yields of medium- to long-dated bonds to increase significantly. Conversely, Subramaniam (2022) also found that during extremely low-rate regimes, yields on short- to medium-dated bonds tend to decline when geopolitical tensions rise, whilst the impact on yields of long-dated bonds is insignificant.

In addition to the above-mentioned work on sovereign bond yields, Sohag et al. (2022) considered the impact of geopolitical risk on the yields of green bonds, and Bouri et al. (2019) the impact of geopolitical risk on corporate Islamic bonds. Like Subramaniam (2022), Sohag et al. (2022) found that the yield response to geopolitical risk is asymmetric and provides evidence that at the 0.10 to 0.40 quantiles, yields on green bonds tend to decline when geopolitical risk increases. On the other hand, in the 0.80 to 0.90 quantiles, yields tend to have a significantly positive relationship with geopolitical risk.

Despite the initial work in this field, no research has been undertaken to test the impact of geopolitical risk on high-quality sovereign bonds that are issued by a government with considerable exposure to the global economic cycle, such as Australia. As mentioned before, the impact of geopolitical risk on these types of bond markets is not clear. Moreover, understanding the reaction of a specific bond market to geopolitical risk will aid in the decision-making of monetary policy makers as well as portfolio managers. The filling of this research gap will consequently provide valuable information.

## 3. Data Discussion

To obtain a holistic understanding of the dynamics between geopolitical risk and the Australian bond market, measures for various dimensions of the Australian bond market are included as dependent variables. To this end, yields on short-, medium- and long-dated Australian sovereign bonds are included, proxied by rolling yields on 1-year, 10-year, and 20-year Australian sovereign bonds, respectively. These yields are obtained from the Thomson Reuters Datastream database which is populated from the Australian Securities Exchange. Term spreads are also included in the study, including the long-short spread, long-medium spread and medium-short spread. This is done based on quantiles

and will provide insights into the impact of geopolitical risk on the yield curve. These spreads represent the difference between the yield on two bond instruments with different maturities at a given point in time and are calculated based on the yields extracted from the Thomson Reuters Datastream database. Finally, quality spreads are also included in the study, including the spread between the Australian and US short-term and medium-term rates. These spreads represent the difference between the yield on two bond instruments issued by different governments, with the same maturity. Yields on US Treasuries are also extracted from the Thomson Reuters Datastream database. These variables will feature as the dependent variables in the various models.

The main explanatory variable in this study is geopolitical risk. To this end, the Geopolitical Risk Index constructed by Caldara and Iacoviello (2022) is used as a proxy for geopolitical risk. This is a news-based index and is constructed by counting the number of words that relate to geopolitical risk each month. This is derived from the leading 11 international newspapers, namely *The Boston Globe*, the *Chicago Tribune*, *The Daily Telegraph*, the *Financial Times*, *The Globe and Mail*, *The Guardian*, the *Los Angeles Times*, *The New York Times*, *The Times*, *The Wall Street Journal* and *The Washington Post* (Caldara and Iacoviello 2022). An advantage of this index is that it captures a broad range of exogenous global uncertainty, including military threats, wars, terror attacks and trade wars (Balcilar et al. 2018). This index has been used as a proxy for geopolitical risk by several other authors, including Balcilar et al. (2018), Bouri et al. (2019), Le and Tran (2021), and Sohag et al. (2022). The index is obtained from Matteoiacoviello (2022), the official Geopolitical Risk Index database.

Guided by the relevant literature, several controlled variables are included in the study. Like Aristei and Martelli (2014), the US VIX index is included to capture international market-related volatility. Furthermore, as suggested by Poghosyan (2014), the Australian government's debt-to-GDP ratio is included to capture fiscal dynamics. A limitation is that Australian government debt figures are released only annually; therefore, linear interpolation is used to approximate monthly data points for the government debt-GDP variable. Furthermore, the AUD/USD exchange rate is included to capture exchange rate dynamics, as suggested by Aristei and Martelli (2014). Given the importance of monetary policy as a determinant of yields, the cash rate set by the RBA is included to capture monetary policy movements. Additionally, in light of the findings by Bianchi et al. (2017), the Australian sovereign credit rating is included. To this end, the rating scale proposed by Cantor and Packer (1996) is used in this study to transform credit ratings into a numerical series, with a numerical scale ranging from 1 to 16, where 1 corresponds with the lower rating, and 16 with the highest. Data on these controlled variables is obtained from the Thomson Reuters Datastream.

All the variables in the study are monthly, including the corresponding credit rating that is recorded monthly. Furthermore, multiple time horizons are used for each model, depending on the first issuance date of the bond instrument under analysis. Variation in the first issuance date of the various bond instruments is a limitation, and therefore this study adopts multiple time horizons. Table 1 shows the time horizons for the models related to the different bond yields and bond spreads considered in this study.

**Table 1.** Time horizons.

| Variable | Time Horizon (MM/YYY) |
|---|---|
| 1-year yield | 01/2000 to 08/2022 |
| 10-year yield | 02/1990 to 08/2022 |
| 20-year yield | 12/2013 to 08/2022 |
| Long-short spread | 12/2013 to 08/2022 |
| Long-medium spread | 12/2013 to 08/2022 |
| Medium-short spread | 2000/01 to 08/2022 |
| Australian-US short-term spread | 2000/01 to 08/2022 |
| Australian-US medium-term spread | 1990/02 to 08/2022 |

Source: self-constructed.

## 4. Methodology Discussion

In this study, quantile regression methodology was employed to determine the asymmetric dynamics between geopolitical risk and Australian sovereign bond yields. Quantile regression analysis allows one to capture any heterogeneity in the relationship across quantiles (Subramaniam 2022). This is a popular approach in literature to capture asymmetric relationships—see, for example, Koenker and Xiao (2006), Bouri et al. (2019), Kannadhasan and Das (2020), and Subramaniam (2022). Quantile regression analysis makes no normality assumptions and is conducive to model variables with outliers (Kannadhasan and Das 2020). This is ideal for research on shocks, given its erratic nature.

The quantile regression framework proposed by Koenker and Xiao (2006) was employed. The model implemented for the various bond yields is specified as follows:

$$Q_{BYxt}(\tau_k|\alpha_i, GPR_t) = \alpha_i(\tau) + \sum_{j=1}^{p} \beta_i(\tau)BY_{xt-j} + \gamma_i(\tau)BY_{xt-1}I(BY_{xt-1}|> BY^q) + \theta_i(\tau)GPR_t + \delta_i(\tau)CV_{ti} + e^t \quad (1)$$

where $BY_{xt}$ denotes the bond yield at month $t$, for the yield related to maturity $x$. It should be noted that this is not a panel setup, and that separate quantile regression models will be estimated for each respective yield measure, $x$. The term $GPR_t$ denotes geopolitical risk at month $t$. The unobserved individual effect is shown by $\alpha_i$ and $\tau$ shows the estimation of a coefficient at the $\tau$th quantile. Furthermore, $\beta_i(\tau)$ is used to control the influence of the autoregressive parameter. The term $\gamma_i(\tau)$ is used to control the influence of extreme yields by using an Indicator Function variable. If the yield exceeds a threshold value of $BY^q$ at month $t-1$, then the Indicator Function, $(BY_{xt-1}|> BY^q)$, takes the value of 1, otherwise 0. The $BY^q$ at the 95th quantile of the uncorrelated distribution is taken. The lag order $p$ is determined by the Bayesian Information Criterion. Furthermore, $\theta_i(\tau)$ shows the impact of $GPR_t$ on $BY_{xt}$ at the $\tau$th quantile. Lastly, $CV_{ti}$ represents a $i$x1 vector of controlled variables, and the term $\delta_i(\tau)$ indicates the impact of each controlled variable on $BY_{xt}$ at $\tau^{th}$ quantile. Using the same notation, the model implemented for the various bond yield spreads, $x$, denoted as $BYS_{xt}$, is specified as follows:

$$Q_{BYSxt}(\tau_k|\alpha_i, GPR_t) = \alpha_i(\tau) + \sum_{j=1}^{p} \beta_i(\tau)BYS_{xt-j} + \gamma_i(\tau)BYS_{xt-1}I(BYS_{xt-1}|> BYS^q) + \theta_i(\tau)GPR_t + \delta_i(\tau)CV_{ti} + e^t \quad (2)$$

A major benefit of quantile regression analysis is that the model allows one to identify asymmetries between variables at different quantiles. In other words, quantile regression analysis allows one to study the relationship among variables at different levels and the assumption of strict linearity is dropped. It speaks for itself that the quantile regression model is only useful if the relationships among variables in the model are asymmetric across quantiles. In accordance with Kovačević (2019), the symmetric quantiles test was employed to test for these asymmetries. If heterogeneity is present, then the use of a

quantile regression analysis is justified. As an initial test for relational asymmetries among variables, the Wald asymmetric test was employed foreach model, in accordance withthe following null hypothesis:

$$H_0 \ : \ \frac{\beta(\tau_j) + \beta(\tau_{K-j-1})}{2} = \beta\left(\frac{1}{2}\right) \qquad (3)$$

For $j = 1, \ldots, \frac{K-1}{2}$.

The assumption here is that the mean value $\tau_{(k+1)/2}$ is equal to 0.5, and the remaining $\tau/$s are symmetric around 0.5.

Koenker and Bassett (1982) propose that the slopes at various quantiles should be tested for differences as a further test of heterogeneity and robustness. To this end, the Wald Slope Equality test was conducted across quantiles to test for heterogeneous slopes. In accordance with Sarkodie, Koenker and Bassett (1982) the hypotheses for this test are expressed as:

$$H_0 \ : \ \beta_1(\tau_1) = \beta_1(\tau_2) = \beta_1(\tau_k)$$

$$H_1 \ : \ \beta_1(\tau_1) \neq \beta_1(\tau_2) \neq \beta_1(\tau_k)$$

$\beta_1$ is the slope coefficient at various quantiles, up to quantile $\tau_k$. In this study, the relationship between geopolitical risk and the various yield measures was evaluated at quantiles 0.1, 0.2, 0.3, 0.4, 0.5, 0.6, 0.7, 0.8, and 0.9. The results and a discussion thereof are presented in the next section. Furthermore, to test if a long-run relationship among the yields under consideration and the corresponding explanatory variables in the model exists, the Engel and Granger cointegration testing procedure was implemented. In this light, the residual of each model was tested for stationarity by employing the Augmented Dickey Fuller unit root test. Furthermore, to test the short-run dynamics of each model, a quantile error correction model was estimated for each of the base models. Equations (1) and (2) can be restated as follows to represent the error correction model:

$$Q_{D(DY;DYS)xt}(\tau_k | \alpha_i, \ D(GPR)_t)$$
$$= \alpha_i(\tau) + \sum_{j=1}^{p} \beta_i(\tau)D(BYS)_{xt-j} + \gamma_i(\tau)D(BYS)_{xt-1}I\big(D(BYS)_{xt-1}\big| > D(BYS)^q\big) \qquad (4)$$
$$+\theta_i(\tau)D(GPR)_t + \delta_i(\tau)D(CV)_{ti} + FC_{t-1} \ + e^t$$

$FC_{t-1}$ is the one period lag of the residual and represents the feedback coefficient that shows how a deviation from the long-run equilibrium corrects in the short run. $D$ depicts a difference in the variable.

## 5. Results and Discussion

Firstly, the results related to the tests for the relational asymmetry among quantiles, the slope equality test and the symmetric quantiles test, depicted in Table 2, are considered. These results indicate if asymmetries in Australian sovereign bond yields and geopolitical risk dynamics exist across quantiles. The presence of asymmetries across quantiles justifies the use of a quantile regression analysis.

The results in Table 1 reveal that, except for the long-short spread series, the null hypothesis of equal slopes across quantiles is rejected, at least at a 95% confidence level, and thus all the other yield measures have slope asymmetries across quantiles. Furthermore, apart from the long-short spread series, the null hypothesis of symmetric quantiles is rejected, at least at a 90% confidence level, and therefore, quantiles are asymmetrical. These results show that a linear estimation of the dynamics between geopolitical risk and Australian bond yields will not be optimal, and that the underlying regime does have an impact on the relationship. Therefore, these results justify the use of quantile regression analysis to test the relationship among various yield measures and geopolitical risk, apart from the long-short spread measure (Koenker and Bassett 1982). The relationship between geopolitical risk and the long-short spread will only be considered at a median level.

**Table 2.** Asymmetry test results.

|  | *p*-Value of Slope Equality Test | *p*-Value of Symmetric Quantiles Test |
|---|---|---|
| Short-term rates | 0.000 *** | 0.039 ** |
| Medium-term rates | 0.000 *** | 0.016 ** |
| Long-term rates | 0.049 ** | 0.093 * |
| Long-short spread | 0.254 | 0.969 |
| Long-medium spread | 0.025 ** | 0.061 * |
| Medium-short spread | 0.015 ** | 0.082 * |
| Short-term risk spread | 0.000 *** | 0.072 * |
| Medium-term risk spread | 0.000 *** | 0.041 ** |

\*, \*\*, and \*\*\* denote statistical significance at a 90%, 95%, and 99% confidence level, respectively, based on *p*-values. Source: self-constructed.

Table 3 depicts the results of the various quantile regression models estimated for each respective yield measure at a median level. Firstly, given that the adjusted R-squared of all three models is at least 0.70, it can be concluded that the models are reasonably specified. Note that the Australian sovereign credit rating remained constant from 2003 and that the long-term bond considered in this study was issued only in 2013. The sovereign credit rating is thus not included as an explanatory variable for any of the models where the dependent variable involves the long-term sovereign yield.

**Table 3.** Aggregate quantile regression results.

|  | Short-Term Rates | Medium-Term Rates | Long-Term Rates | Long-Short Term Spread | Long-Medium Term Spread | Medium-Short Term Spread | Short-Term Risk Spread | Medium-Term Risk Spread |
|---|---|---|---|---|---|---|---|---|
| AUD/USD | −0.438 ** | 1.269 *** | 1.819 *** | 6.736 *** | 0.514 ** | 0.436 | 5.109 *** | 3.970 *** |
| Credit rating | −0.125 * | −1.516 *** | N/A | N/A | N/A | −0.757 *** | −1.015 * | −0.353 ** |
| Debt/GDP | 0.038 * | −0.046 ** | 0.092 *** | 0.046 * | −0.0071 | −0.057 *** | −0.130 ** | −0.011 |
| Geo Risk | 0.008 | 0.052 ** | 0.029 *** | 0.001 | −0.040 ** | 0.017 * | 0.088 *** | 0.046 *** |
| RBA policy rate | 0.978 *** | 0.541 *** | 0.977 ** | −0.285 ** | −0.084 ** | −0.444 *** | −0.197 * | 0.183 *** |
| VIX | −0.015 ** | −0.018 ** | −0.012 ** | 0.072 ** | 0.027 ** | 0.012 *** | 0.029 *** | 0.014 *** |
| C | 2.604 * | 25.706 *** | −4.034 ** | 0.418 *** | 0.436 ** | 0.400 *** | 2.851 ** | 2.202 * |
| Adjusted R-squared | 0.839 | 0.750 | 0.781 | 0.71 | 0.703 | 0.772 | 0.785 | 0.718 |
| Co-integration test results | 0.081 * | 0.000 *** | 0.001 *** | 0.004 *** | 0.026 ** | 0.000 *** | 0.002 *** | 0.000 *** |
| ECM coefficient | −0.126 ** | −0.049 * | −0.131 * | −0.084 | −0.117 *** | −0.140 *** | −0.052 * | −0.061 ** |

\*, \*\*, and \*\*\* denote statistical significance at a 90%, 95% and 99% confidence level, respectively, based on *p*-values. Source: self-constructed.

The co-integration results demonstrate that the residuals of all the models are stationary at level, and therefore, a long-run relationship between each yield measure and the corresponding explanatory variables does exist. The results indicate that geopolitical risk does not have a statistically significant impact on short-term sovereign bond yields or long-short term spreads. On the other hand, the results provide evidence that geopolitical risk does have a statistically significant positive impact on medium- and long-term bond yields. Thus, a rise in geopolitical risk typically causes a statistically significant selloff of Australian government bonds and, thus, causes yields to increase. Furthermore, the results provide evidence that geopolitical risk does have a statistically significant impact on long-medium-term spreads and medium-short term spreads. However, geopolitical risk has a negative impact on long-medium-term spreads, and a positive impact on medium-short

term spreads. Additionally, the results illustrate that geopolitical risk has a statistically significant positive impact on both the short- and medium-term risk spread.

Considering the short-run dynamics of each model, the adjustment coefficients of all the models, with the exception of the long-short term spread, are statistically significant at a 90% or higher confidence level. Therefore, unlike the variables in the other models, long-short term spread does not have a statistically significant short-run relationship with the corresponding variables. Further, all the feedback coefficients are negative, indicating that a statistically significant short-term relationship exists among the yield measure in each model, except the long-short-term spread model, and the corresponding explanatory variables. Therefore, a short-term deviation in the long-run equilibrium will significantly adjust and correct in the short-run.

To analyse these dynamics in more detail, the results on the relationship between geopolitical risk and each yield measure at the 10th to 90th quantiles are considered and are presented in Table 4. It is noteworthy that the quantile interval [0.10–0.20] represents an extremely low-rate regime, [0.20–0.40] represents a low-rate regime, [0.40–0.70] represents rates around the mean, and [0.70–0.90] represents a high-rate regime. It should also be noted that the quantile results for short-term yields are also considered, even though the results in Table 3 suggest that geopolitical risk does not have a statistically significant impact on short-term yields at an aggregate level. This is due to the results in Table 2 providing evidence of quantile asymmetries between geopolitical risk and short-term yields. Therefore, geopolitical risk might still have a statistically significant impact at extreme quantiles, which is worth considering. On the other hand, the slope equality test and symmetric quantiles test summarised in Table 2 provide no evidence of quantile asymmetries related to the dynamics between the long-short rate spread and geopolitical risk. The impact of geopolitical risk on the long-short spread will therefore not be considered further.

**Table 4.** Results at quantile level for geopolitical risk.

| | Short-Term Rates | Medium-Term Rates | Long-Term Rates | Long-Medium Term Spread | Medium-Short Term Spread | Short-Term Risk Spread | Medium-Term Risk Spread |
|---|---|---|---|---|---|---|---|
| 0.1 | −0.098 *** | 0.017 *** | 0.027 | −0.012 *** | 0.054 ** | 0.042 *** | 0.03 *** |
| 0.2 | −0.070 * | 0.041 *** | 0.022 *** | −0.019 *** | 0.043 * | 0.038 *** | 0.040 *** |
| 0.3 | 0.004 | 0.037 ** | 0.015 *** | −0.023 *** | 0.010 | 0.006 | 0.046 *** |
| 0.4 | 0.002 | 0.045 ** | 0.036 *** | −0.024 *** | 0.017 | 0.007 | 0.055 *** |
| 0.5 | 0.008 | 0.052 ** | 0.029 *** | −0.040 *** | 0.017 * | 0.088 *** | 0.046 *** |
| 0.6 | 0.005 | 0.057 ** | 0.045 *** | −0.035 *** | 0.015 ** | 0.021 *** | 0.041 *** |
| 0.7 | 0.002 | 0.061 * | 0.031 *** | −0.022 *** | 0.016 | 0.019 ** | 0.037 *** |
| 0.8 | 0.031 *** | 0.053 ** | 0.015 | −0.012 *** | 0.019 | 0.007 | 0.025 * |
| 0.9 | 0.006 | 0.003 | 0.007 | 0.004 | 0.009 | 0.001 | 0.001 |

\*, \*\*, and \*\*\* denote statistical significance at a 90%, 95%, and 99% confidence level, respectively, based on *p*-values. Source: self-constructed.

The results in Table 4 provide evidence that geopolitical risk does have a statistically significant negative impact on short-term bond yields at the 10th and 20th quantiles, and that it has a statistically significant positive impact at the 80th quantile. However, geopolitical risk proves to have a statistically insignificant impact on short-term yields at the other quantiles. Literature, such as the work by Bianchi et al. (2017) and Gupta et al. (2021), provides evidence that yields on short-term sovereign bonds are largely controlled by the central bank, and less sensitive to other market conditions. The results corroborate the fact that investors in short-term Australian sovereign bonds do not expect a significant shift in central bank policy due to increased geopolitical risk during moderate rate regimes, that is, at the 30th to 70th quantiles, and this is in line with the findings by Gupta et al. (2021).

However, this is not the case at either the lower or upper extremes. When rates are at the lower extremes, that is, at the 10th and 20th quantiles, rates tend to decline even further in the face of geopolitical risk, alluding to the expectation of further easing in central bank policy. Contrarily, when rates are high, that is, at a time when there are inflationary pressures, geopolitical risk tends to result in even higher bond yields, thus alluding to expectations of further inflationary pressures.

The results show that geopolitical risk has a statistically significant positive impact on medium-term yields at all the quantiles, except the 90th quantile. It is also interesting to note that there is a general increase in the beta coefficients between the quantiles. This indicates that the magnitude of the impact on yields increases at relatively higher regime levels. This is largely in line with Subramaniam (2022), also finding that the impact at extremely low-rate regimes is insignificant. Yet, this study provides evidence that it is significant at extremely low rates in the Australian bond market.

The results also signal that geopolitical risk has a statistically significant positive impact on long-term yields at the 20th to 70th quantiles, but proves to be insignificant at extreme regimes, that is, at the 10th, 80th and 90th quantiles. These results indicate that both the null and alternative hypotheses in the first hypothesis statement applies, depending on the yield-to-maturity and the prevailing rate regime. As reviewed in the literature, expectations about the impact of geopolitical risk on medium- and long-term Australian bond yields are not clear, because there are potentially two opposing forces at work. On the one hand, highly rated Australian sovereign bonds could act as a safe haven during geopolitical turbulence. On the other hand, geopolitical risk could increase expectations of a global economic slowdown, which could be particularly negative for the exporting Australian economy. The significant increase in yields, signalling an increase in the risk premium, provides evidence that the latter force is the dominant force in the Australian bond market, which aligns with the results found by Subramaniam (2022). The results thus intimate that medium- and long-term Australian sovereign bonds at most of the rate regimes do not act as a safe-haven asset during relatively higher levels of geopolitical risk, despite the very high quality of these assets. This is significant to portfolio managers, as it impacts the inclusion of Australian sovereign bonds as a hedging tool against geopolitical risk. In this light, portfolio managers should rather consider short-dated Australian bonds as a hedge, especially during low-rate regime periods.

The results additionally provide evidence that geopolitical risk does have a statistically significant negative impact on the long-medium term spread at all quantiles apart from the 90th quantile. This shows that the longer end of the yield curve tends to flatten during periods of high geopolitical tension. It is interesting to note that from the 10th up to the 60th quantile, the magnitude of the impact increases per quantile, and starts to decline at the 70th quantile. Therefore, in this case, the null hypothesis of the second hypothesis statement should be accepted.

Monetary policy makers should take note of the increase in yield curve sensitivity to geopolitical risk across the above-mentioned quantiles. This should be considered in light of literature proposing that a flatter yield curve at the long end signals expectations of a recession (Umar et al. 2021). As suggested by the results on short-term rates, monetary policy makers typically reduce rates during extremely low-rate regimes in the face of increased geopolitical risk, but largely leave rates unchanged in moderate- and high-rate regimes. Yet, the results suggest that monetary policy makers should consider the flattening of the yield curve across the majority of the spread regimes as a signal to start easing policy at all the rate regimes except at the 90th quantile, thereby pre-empting a slowdown in inflation to remain at the head of the curve.

Furthermore, the results show that geopolitical risk has a statistically significant positive impact on the medium-short term spread at the 10th, 20th, 50th and 60th quantiles, but is insignificant at the other quantiles. This suggests that the middle of the curve tends to steepen during periods of high geopolitical tension, but only at an extremely low-rate regime, and at a normal rate regime. Given that the impact of geopolitical risk on short-term

yields is limited, as evidenced by the results, this spread is largely driven by movements in medium-term yields. It thus indicates that investors require a relatively higher risk premium to invest in medium-term Australian sovereign bonds, relative to short-term bonds, during periods of increased geopolitical risk. Therefore, in this case, the alternative hypothesis of the second hypothesis statement should be accepted.

Regarding the quality spreads, the results show that geopolitical risk has a statistically significant positive impact on the short-term quality spread at extremely low spread quantiles, that is, the 10th and 20th quantiles, as well as at the middle to upper quantiles, that is, the 50th to 70th quantiles. However, geopolitical risk proves to be insignificant at extreme spread levels, that is, the 80th and 90th quantiles. Thus, at levels where the bond market is already priced at high levels of risk, the additional geopolitical risk does not further impact the spread. The impact of geopolitical risk on medium-term quality spreads has a larger breadth across quantiles and reveals a statistically significant positive impact from the 10th to the 90th quantiles. Therefore, in this case, the null hypothesis of the third hypothesis statement should be accepted.

This is in line with literature providing evidence that money flows into US sovereign bonds during periods of uncertainty (Afonso et al. 2015; Aristei and Martelli 2014). These results advance that, by and large, both short- and medium-term sovereign bonds issued by the US government outperform the short- and medium-term sovereign bonds issued by the Australian government. Portfolio managers should thus rather consider US sovereign bonds as a hedge against geopolitical risk, despite the higher credit rating allocated to Australian sovereign bonds. The results also signal that geopolitical risk has an asymmetric impact on yield measures at different rate regimes, and an asymmetric impact on yield spreads at different spread levels. It is thus suggested that policy makers should not consider geopolitical risk as a homogeneous risk but should formulate policy with the identified asymmetries in mind.

## 6. Conclusions

The main aim of this study was to determine the dynamics between Australian sovereign bond yields and geopolitical risk. Focusing on the sovereign bond market of a single country, rather than on those of a panel of countries, afforded this study the opportunity to obtain a detailed and holistic view of the dynamics between a change in geopolitical risk and various aspects of the Australian sovereign bond market. To achieve the main aim of the study, quantile regression analysis was utilised to model the data, and the Geopolitical Risk Index created by Caldara and Iacoviello (2022) was used as a proxy for geopolitical risk. Furthermore, a range of sovereign yield measures, including short- to long-term yields, term spreads and quality spreads, were considered in the study.

The study offers several key findings. Firstly, the study provides evidence that geopolitical risk has an asymmetric impact on yields at different rate regimes, and an asymmetric impact on yield spreads at different spread levels. Secondly, an increase in geopolitical risk only impacts short-term yields at extreme regimes, that is, the 10th, 20th and 80th quantiles; however, the impact is, by and large, insignificant. At the 10th and 20th quantiles, the impact on yields is typically negative, but it is positive at the 80th quantile. Thirdly, an increase in geopolitical risk does have a statistically significant positive impact on medium- and long-term yields across most quantiles. This indicates that an additional risk premium is required for medium- and long-term Australian sovereign bonds during periods of increased geopolitical risk. The main implication of this from a fiscal policy perspective is that the cost of newly issued government debt will increase, and re-financing of government debt will occur at a higher rate.

Furthermore, an increase in medium to long-term sovereign yields could have a negative impact on all three core mandates of the RBA. It could cause price instability since an outflux of capital out of sovereign debt instruments could result in a depreciation of the Australian dollar. This would particularly be the case if capital flows out of the country to other debt instruments that are perceived to be safe havens. Moreover, an increase in

the cost of debt could result in the need to cut other government expenses which, in turn, could cause an economic slowdown and an increase in unemployment. This will impact the RBA's full employment and stable economic growth mandate. Given this information, monetary policy makers at the RBA should consider increasing their balance sheet during these periods by means of purchasing medium- and long-term sovereign debt as part of their open market operations. During times of selling, the central bank would reduce the risk premium. Form a portfolio management perspective, the results suggest that medium- and long-term Australian bonds might not act as a good hedge against geopolitical risk, and portfolio managers should rather consider short-dated Australian bonds as a hedge, especially during low-rate regimes

Fourthly, an increase in geopolitical risk tends to result in a steeper yield curve at the belly of the curve at extremely low and middle quantiles, but causes the yield curve at the middle to long end to flatten across all the quantiles, except the 90th quantile. A flatter yield curve at the medium to long end typically signals a slowdown in the economy and relates to the stable economic growth and full employment mandate of the RBA. The RBA could consider this signal to pre-empt a slowdown and could reduce short-end rates to limit the real economic impact of an increase in geopolitical risk. Furthermore, to reduce the flattening of the curve, the RBA could increase their buying of medium-term sovereign bonds relative to long-term bonds. Lastly, medium-term quality spreads tend to significantly increase at all the quantiles. This points towards the fact that funds tend to flow out of medium-term Australian sovereign bonds and US medium-term bonds tend to attract funds during periods of increased geopolitical risk. Once again, the RBA could reduce this spread by means of purchasing medium-term Australian sovereign bonds.

These results provide insights into the dynamics between sovereign risk and Australian bond market dynamics. Given the important role that sovereign bonds play in local economic activity, the findings reached in this study could aid Australian monetary policy makers in formulating policy that will aid in reducing the impact of geopolitical risk on the economy. The findings also provide insights into the hedging capacity of Australian sovereign bonds against geopolitical risk.

The main limitation of this study includes variation in the first issuance date of the various bond instruments considered in the study. This necessitated the adoption of multiple time horizons. Two main delimitations shaped the focus of this study. Firstly, the study focused only on the Australian bond market and given the heterogenous nature of the dynamics between geopolitical risk and bondyields, results might differ if the same approach is applied to different markets. Secondly, the study focused only on sovereign bond yields, notwithstanding that corporate bond yields also play a key role in the capital market.

Given these delimitations, the field would benefit from more studies on the dynamics between sovereign bond yields and geopolitical risk in other markets. In this light, future studies could use this study along with the other initial studies as a base, and test if the same dynamics exist in other markets, and if not, how they differ. This would allow for tailored monetary policy making. Additionally, the field would benefit from studies on the dynamics between geopolitical risk and corporate bond yields. This field could also expand into qualitative studies, such as studies that consider the perspectives of bond market participants during periods of geopolitical tensions. This would provide an alternate angle to the problem and would provide detailed reasoning behind certain investment decisions in the face of geopolitical tensions. Lastly, studies seeking other forms of hedging options against geopolitical risk will be beneficial for the portfolio management industry.

**Funding:** This research received no external funding.

**Data Availability Statement:** Data available in a publicly accessible repository that does not issue DOIs. Data could be obtained through numerous databases such as Reuters DataStream: https://www.refinitiv.com/en/products/datastream-macroeconomic-analysis?utm_content=sitelink&utm_medium=cpc&utm_source=google&utm_campaign=596226_PaidSearchInvestmentSolutionsBAU&elqC

ampaignId=16987&gclid=CjwKCAiA9NGfBhBvEiwAq5vSy4Pw4x7dnt5lq4Arz4wV9jr_14aEBgYszi Lwr3If5iOWmOuV7tsJLxoCx9MQAvD_BwE&gclsrc=aw.ds (accessed on 13 February 2023). The Geopolitical risk index can be obtained by following this link: https://www.matteoiacoviello.com/g pr.htm (accessed on 13 February 2023).

**Conflicts of Interest:** The author declares no conflict of interest.

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
