# Peer review of "Geopolitical Risks and Yield Dynamics in the Australian Sovereign Bond Market"

_jrfm, doi:10.3390/jrfm16030144_

Round 1

Reviewer 1 Report

The paper examines geopolitical risks and yield dynamics in Australian sovereign bond market and reported that increase in geopolitical risk only impacts short term yield at extreme regime.  

Grammar/language check – see line 131, Firstly, this is the first study that …………… [the author needs a language editor to reassess the paper].

Abstract – The abstract look promising but policy implications must be strengthened. 

Introduction – The introduction is poorly written, and contribution of the studies and hypotheses tested are not specified.  

Literature review – The author needs to redraft the literature review and identify the gaps in the existing literature.  

Data – The data should be clearly separated from the methodology section. The sources of data, scope covered, and descriptions should be clear. The author should clarify if monthly, quarterly, or yearly data are used.

Methodology – The author should check the appropriateness of the methodology and whether the approach answer questions on the short and long run dynamic among the variables. 

Conclusion - The policy implication is omitted, no limitation of study as well as suggestion for future research.

Author Response

Reviewer 1

Reviewer Comment

Correction/author response

Grammar/language check – see line 131, Firstly, this is the first study that …………… [the author needs a language editor to reassess the paper].

The article was submitted for professional language editing by a professional English editor.

Abstract – The abstract look promising but policy implications must be strengthened. 

The last part of the abstract is expanded to state how the results of this study have implications for monetary policymakers, fiscal policymakers, and portfolio managers.

Introduction – The introduction is poorly written, and contribution of the studies and hypotheses tested are not specified.  

The introduction was restructured. Furthermore, the three main hypotheses tested in the study, as well as the contribution of the study were added at the end of the introduction section.

Literature review – The author needs to redraft the literature review and identify the gaps in the existing literature.  

A number of paragraphs were added at the end of the literature to clearly show the current work in the field, and thereby depict the research gap filled by this paper.

Data – The data should be clearly separated from the methodology section. The sources of data, scope covered, and descriptions should be clear. The author should clarify if monthly, quarterly, or yearly data are used.

The data discussion and methodology discussion are now separate (Sections 3 and 4).

The sources of the variables in the study are added in the data discussion section and the frequency of the variables is depicted.

Methodology – The author should check the appropriateness of the methodology and whether the approach answers questions on the short and long run dynamic among the variables. 

To address this comment, a co-integration analysis is added to test for a long-run relationship between the variables in each model. Additionally, to test the short-term dynamics within each model, an Error Correction methodology (ECM) was employed, based on each of the yield variables in the study. This shows how short-term deviations from the long-run equilibrium corrects.

Conclusion - The policy implication is omitted, no limitation of study as well as suggestion for future research.

Policy implications are added to the conclusion, along with recommendations to policymakers.

Furthermore, a section on the limitation and delimitations of the study is added. Lastly, suggestions for future research are discussed at the end of the conclusion.

Reviewer 2 Report

Title: Geopolitical risks and yield dynamics in the Australian sovereign bond market.

I appreciate the chance to serve as a reviewer on this paper. The paper is well written and suits the scope of the issue. I believe that the paper could be of interest to an international audience since it deals with a very interesting topic. I recommend accepting the paper after minor revisions, so I urge the author(s) to consider the following comments and improve the paper accordingly.

1.      It would be useful if the authors explain in detail how the market regulators benefit by their results.

2.      It would be useful if the authors explain how their model catch up asymmetries. (see for instance Floros et al. 2020).

3.      What will be the economic implications in macro – level in business activity?

4.      It would also be useful for the audience and future researchers if a guide for the future research is provided: how this research could be used concretely to open new pathways? Is it possible to provide some examples and possible directions for future research? 

This is a good work and I think that a revised version with the abovementioned concerns could be a contribution to the literature.

Literature

Floros C., Gkillas K., Konstantatos C., Tsagkanos A., (2020) “Realized measures to explain volatility changes over time” Journal of Risk Financial Management. Vol 13(6), 125-144

Author Response

Reviewer 2

 It would be useful if the authors explain in detail how the market regulators benefit from their results.

This comment is addressed in the introduction of the article, specifically at the end of the introduction. This is done by discussing how each yield measure analysed in the study relates to one or more of the core mandates of the Australian Reserve Bank.

 It would be useful if the authors explain how their model catch up with asymmetries. (see for instance Floros et al. 2020).

A section under the methodology discussion is added to explain how asymmetries are shown by a Quantile regression analysis. Furthermore, the Wald hypothesis test for quantile asymmetries is discussed as part of the methodology section.  

 What will be the economic implications at macro – level in business activity?

This comment is mainly addressed in the conclusion of the article, where the implications of the results are linked to economic activity and policy recommendations are provided based on the implications. This is also touched on in the discussion of results.

 It would also be useful for the audience and future researchers if a guide for the future research is provided: how this research could be used concretely to open new pathways? Is it possible to provide some examples and possible directions for future research? 

This comment is mainly addressed in the conclusion where suggestions for future research are added.

Round 2

Reviewer 1 Report

Table 2 need to be revisited.

Author Response

Reviewer comment

Author response

Table 2 needs to be revisited.

The table is revisited, and two adjustments are made:

1.       The footnote is adjusted to include the 90% significance denotation.

2.       The explanation of the results depicted in the table is extended.
